# Generative Human Trajectory Recovery via Embedding-Space Conditional Diffusion

Kaijun Liu [1 2]   Sijie Ruan [3]   Liang Zhang [2]   Cheng Long [2]   Shuliang Wang [3]   Liang Yu [4]

## Abstract

Recovering human trajectories from incomplete or missing data is crucial for many mobility-based urban applications, e.g., urban planning, transportation, and location-based services. Existing methods mainly rely on recurrent neural networks or attention mechanisms. Though promising, they encounter limitations in capturing complex spatial-temporal dependencies in low-sampling trajectories. Recently, diffusion models show potential in content generation. However, most of proposed methods are used to generate contents in continuous numerical representations, which cannot be directly adapted to the human location trajectory recovery. In this paper, we introduce a conditional diffusion-based trajectory recovery method, namely, DiffMove. It first transforms locations in trajectories into the embedding space, in which the embedding denoising is performed, and then missing locations are recovered by an embedding decoder. DiffMove not only improves accuracy by introducing high-quality generative methods in the trajectory recovery, but also carefully models the transition, periodicity, and temporal patterns in human mobility. Extensive experiments based on two representative real-world mobility datasets are conducted, and the results show significant improvements (an average of 11% in recall) over the best baselines.

## 1. Introduction

Mobility data plays a prominent role in many urban applications, e.g, next location recommendations (Feng et al., 2018;

Han et al., 2025), epidemic prevention (Tang et al., 2023) and urban planning (Yuan et al., 2014). However, due to privacy concerns or device malfunctions, users may not report their locations to the service provider continuously, which makes human trajectories sparsely distributed in space and unevenly observed in time, and thus affects the effectiveness of downstream applications. Accurate recovery of missing trajectory points is also vital for related trajectory-modeling tasks; for example, on-road trajectory modeling frameworks such as MMTraj+ (Liu et al., 2025) rely on complete sequence inputs to learn trajectory patterns and destination cues via multi-task learning. Recovered human trajectories, especially those involving Points of Interest (POIs), can greatly benefit downstream applications such as POI recommendation, targeted advertising, and ride-sharing services by enabling more personalized and context-aware decisions. For this reason, human trajectory recovery, which infers human trajectories at a fine-grained level, raised more and more attention recently.

Existing human trajectory recovery work leverages Recurrent Neural Network (Liu et al., 2016; Wang et al., 2019) or attention mechanism (Xia et al., 2021) for capturing the spatial-temporal dependencies and resort sparse time interval encoding modules for handling unevenly observed trajectory records. Recent studies (Xia et al., 2021; Sun et al., 2021; Deng et al., 2023) further found that explicitly utilizing historical trajectory can enhance the performance due to the strong periodicity nature of the human trajectories. However, these approaches face significant limitations in handling other key characteristics of human mobility. First, they struggle to capture intricate spatial-temporal dependencies - the interplay between spatial relationships (proximity and spatial transitions between locations) and temporal patterns (sequential dependencies or periodicity of behaviors in historical trajectory). Second, it is difficult for modeling uncertainty caused by sparse or incomplete observations, which is common in real-world mobility data. Most deterministic methods produce a single imputation sample, potentially oversimplifying the variability in human behavior.

Although existing methods achieve appealing performance, these approaches all made the traditional predictive recov-

[1]Alibaba-NTU Singapore Joint Research Institute, Singapore [2]College of Computing and Data Science, Nanyang Technological University, Singapore [3]School of Computer Science and Technology, Beijing Institute of Technology, China [4]Alibaba Cloud, Hangzhou, Zhejiang, China. Correspondence to: Sijie Ruan <sjruan@bit.edu.cn>.

*Proceedings of the 42$^{nd}$ International Conference on Machine Learning*, Vancouver, Canada. PMLR 267, 2025. Copyright 2025 by the author(s).

ery, which has above limitations in complex and uncertain scenarios inherently in human mobility. Consequently, the recovery accuracy and scenarios of downstream applications are limited. For instance, a person may follow some routines from home to office daily but occasionally he/she may follow different routes or change his/her preference. In such scenarios, traditional methods typically provide a biased deterministic imputed trajectory. However, with a generative approach to inference, a set of imputed trajectory locations can be generated through sampling or various averaging techniques on imputation samples.

To address these fundamental limitations, we leverage diffusion models, which have shown superior performance in many tasks against other generative models, e.g., image generation (Ho et al., 2020) and audio synthesis (Kong et al., 2021). Furthermore, conditional diffusion models are recently developed for the time-series imputation (Tashiro et al., 2021) given observed entries as input, which inspires us to design a trajectory recovery model in the conditional diffusion manner.

However, it is non-trivial to apply the conditional diffusion model to solve the trajectory recovery problem due to two issues. *Firstly*, the imputation targets of conventional diffusion models are continuous numerical values, which can be directly obtained via the denoising process, while those in trajectory recovery are discrete ID-represented locations (e.g., POI locations or geographical grid cells) - in this case, the transition and periodicity patterns of human trajectories are required to be fully exploited. *Secondly*, the model must simultaneously consider those abovementioned limitations in handling key characteristics of human mobility - both temporal dependencies and the complex spatial relationships between current and historical trajectories during the denoising process.

To tackle these issues, we propose a novel conditional diffusion model for human trajectory recovery, namely, DiffMove. Our model incorporates these specialized components: 1) a novel embedding-based (with encoding and decoding) conditional diffusion framework that handles discrete locations while preserving spatial relationships, 2) a Spatial Conditional Block equipped with diffusion-oriented graph neural network and attention mechanism, which captures the sparse spatial transition patterns and periodicity temporal patterns from the current trajectories and historical trajectories. 3) a Target Conditional Block that effectively utilizes historical information despite sparse sampling. 4) a Denoising Network Block to handle uncertainty. Our contributions are three-fold:

- We propose a trajectory recovery framework DiffMove[1],

---
[1]The code of this paper will be released in the link https://github.com/KaijunL/DiffMove

which provides a solution to impute discrete locations leveraging diffusion models by performing the denoising process in embedding space and decoding the inferred embeddings back to discrete locations. To the authors' knowledge, we are the first to design spatial temporal conditional diffusion models for human trajectory recovery task.

- We design Spatial Conditional Block, Target Conditional Block and Denoising Network Block to fully fuse the knowledge of the current trajectories and historical trajectories during the conditional diffusion process and tackle the above challenges.

- Extensive experiments on two real-world mobility datasets demonstrate that DiffMove significantly outperforms state-of-the-art baselines, achieving an average improvement of 11% in Recall.

## 2. Related works

**Human Trajectory Recovery:** The human trajectory recovery problem we address focuses on free-space settings, unlike MTrajRec (Ren et al., 2021) and RNTrajRec (Chen et al., 2023b), which focus on vehicles' trajectories constrained by road networks (road segments). Human trajectory recovery can be categorized into two types: the former treats missing locations in trajectories as continuous two dimensional values, i.e., latitude and longitude, to be imputed (Alwan & Roberts, 1988; Moritz & Bartz-Beielstein, 2017; Wang et al., 2019), while the latter infers locations from a discrete candidate location pool (Liu et al., 2016; Xia et al., 2021). The former is suitable to recover trajectories with high sampling frequency, e.g., vehicle trajectories, where the local context plays a more important role for the imputation, while the latter is more feasible for highly sparse trajectories, e.g., human trajectories, where the transition and periodicity dependency modeling are the main focus. In human trajectory recovery, the de-facto approach is to explicitly utilize historical trajectory when imputating the current trajectory. For example, AttnMove (Xia et al., 2021) utilizes a multi-stage attention mechanism to recover missing locations. PeriodicMove (Sun et al., 2021) constructs day-level graphs to model complex transition patterns among locations. TRILL (Deng et al., 2023) is a trajectory recovery model utilizing graph convolutional networks, combining global and local mobility patterns. Chen et al. (2023a) proposes a framework called TERI, to tackle trajectory recovery in a two-stage process, with a different problem setting focusing on addressing the special cases of irregular time interval. Ours is more addressing on the generative manner using diffusion model to solve the regular trajectory recovery problem. Existing human trajectory recovery work imputes missing locations in a deterministic manner, which omits the uncertain nature of trajectories and thus constrains the recovery accuracy and scenarios

of downstream applications. In addition, the relationship between locations to be imputed and historical trajectories are not well-modeled.

**Diffusion Model for Temporal and Spatio-temporal Data:** Diffusion models have found extensive applications in tasks related to time series and spatio-temporal data generation (Wang et al., 2024), imputation (Tashiro et al., 2021), and forecasting (Wen et al., 2023) due to their competence in modeling high-dimensional data distributions. From the time series generation aspect, diffusion models can be used for the synthesis of electronic health records (EHR) (Alcaraz & Strodthoff, 2023; He et al., 2023; Yuan et al., 2023). Many of these studies adopt the denoising network architecture initially proposed in DiffWave (Kong et al., 2021), which utilizes bidirectional dilated convolution to capture correlations between different time steps. CSDI (Tashiro et al., 2021) leverages diffusion models for probabilistic time series imputation, i.e., generating missing values conditioned on observed data points. DiffTraj (Zhu et al., 2024) represents the first attempt to generate GPS trajectories using an unconditioned diffusion probabilistic model. However, it focuses on generating task of raw GPS data in continuous space instead of discrete sparse locations that human trajectories always involve. TrajGDM (Chu et al., 2023) employs a diffusion model to capture universal mobility patterns, for trajectory generation, but it focuses on simulating synthetic human mobility instead of recovery task on current trajectory. A recent work DiffSTG (Wen et al., 2023) studied the spatial-temporal graph forecasting problem and introduced a denoising network UGnet, which is capable of capturing spatial-temporal dependencies among various geographical locations. However, DiffSTG focuses on predicting numerical readings of geographical sensors in different locations across different time, while we focus on recovering discrete locations in human trajectories.

## 3. Preliminaries

### 3.1. Problem Statement

**Definition 1** (**Trajectory**). *The trajectory is a chronological sequence of a user's locations within a single day. Let $\mathcal{T}_u^j : l_u^{j,1} \to l_u^{j,2} ... \to l_u^{j,k} ... \to l_u^{j,K}$ represent the trajectory of user $u$ on the $j$-th day, where $l_u^{j,k}$ denotes the visited location during the $k$-th time slot within a specified time interval. If the location for the $k$-th time slot is not observed, $l_u^{j,k}$ is marked as null, i.e., $l_u^{j,k}$ is missing.*

**Definition 2** (**Current and Historical Trajectory**). *For a given targeted day $J$ and user $u$'s trajectory $\mathcal{T}_u^J$, we define $\mathcal{T}_u^J$ as the user's current trajectory, while the historical trajectories comprise $u$'s trajectories in the past $(J-1)$ days, denoted as $\{\mathcal{T}_u^1, \mathcal{T}_u^2, ..., \mathcal{T}_u^{J-1}\}$.*

We follow (Xia et al., 2021) to formulate the human trajectory recovery problem as follows:

**Problem Definition**. *Given user $u$'s trajectory $\mathcal{T}_u^J$ along*

with historical trajectories $\mathcal{T}_u^1, \mathcal{T}_u^2, ..., \mathcal{T}_u^{J-1}$, *the task is to recover the missing locations, i.e., $\forall$ null in $\mathcal{T}_u^J$, thereby reconstructing the complete trajectory for the current day.*

### 3.2. Denoising Diffusion Probabilistic Model

Denoising Diffusion Probabilistic Models (DDPM) (Ho et al., 2020) are deep generative models, which map data from the normal distribution to another distribution via a learnable denoising network step by step so that we can easily generate a data sample following the similar distribution of $q(\boldsymbol{x}_0)$ by sampling a random Gaussian noise. DDPM is composed of a forward process and a reverse process.

In the forward process, Gaussian noise is gradually added to the data sample $\boldsymbol{x}_0 \sim q(\boldsymbol{x}_0)$ by a Markov chain. A closed form exists to transform the initial data sample $\boldsymbol{x}_0$ to the data sample $\boldsymbol{x}_t$ at arbitrary time step $t$ by the reparameterization trick: $\boldsymbol{x}_t = \sqrt{\bar{\alpha}_t}\boldsymbol{x}_0 + \sqrt{1 - \bar{\alpha}_t}\boldsymbol{\epsilon}$   (1)

where $\bar{\alpha}_t = \alpha_1 \alpha_2 \ldots \alpha_t$, $\alpha_t = 1 - \beta_t$, $\beta_t \in (0, 1)$ denotes the noise level and $\boldsymbol{\epsilon}$ is sampled from a Gaussian noise $\mathcal{N}(\boldsymbol{0}, \boldsymbol{I})$.

The reverse process iteratively denoises a pure Gaussian noise $\boldsymbol{x}_T \sim \mathcal{N}(\boldsymbol{0}, \boldsymbol{I})$ to generate the data sample $\boldsymbol{x}_0$ following the similar distribution of $q(\boldsymbol{x}_0)$. The transformation between data of two consecutive steps can be formulated as follows: $p_\theta(\boldsymbol{x}_{t-1}|\boldsymbol{x}_t) = \mathcal{N}(\boldsymbol{x}_{t-1}; \boldsymbol{\mu}_\theta(\boldsymbol{x}_t, t), \sigma_\theta(\boldsymbol{x}_t, t)\boldsymbol{I})$   (2)

where $\theta$ is shared among different denoising time steps. The parameters of $p_\theta(\boldsymbol{x}_{t-1}|\boldsymbol{x}_t)$ are calculated as follows: $\boldsymbol{\mu}_\theta(\boldsymbol{x}_t, t) = \frac{1}{\bar{\alpha}_t}(\boldsymbol{x}_t - \frac{\beta_t}{\sqrt{1 - \bar{\alpha}_t}}\boldsymbol{\epsilon}_\theta(\boldsymbol{x}_t, t))$, $\sigma_\theta^2(\boldsymbol{x}_t, t) = \frac{1 - \bar{\alpha}_{t-1}}{1 - \bar{\alpha}_t}\beta_t$   (3)

where $\boldsymbol{\epsilon}_\theta$ is the denoising network, which takes the noise-added data $\boldsymbol{x}_t$ and the time step $t$ as inputs and produces the predicted noise. By iteratively sampling according to Eq. (2), the generated data $\hat{\boldsymbol{x}}_0$ is finally obtained. During the training stage, the denoising network parameters $\theta$ can be learned by minimizing $L(\theta) = \mathbb{E}||\boldsymbol{\epsilon} - \boldsymbol{\epsilon}_\theta(\boldsymbol{x}_t, t)||_2^2$, where $\boldsymbol{x}_t$ can be obtained given $\boldsymbol{x}_0$ based on Eq. (1).

## 4. Methodology

The main idea of diffusion-based trajectory recovery is to first transform discrete locations in trajectories into the dense embedding space, then generate the recovered location embeddings via a diffusion model, and finally rebuild the missing locations by an embedding matching process.

To generate satisfactory location embeddings, the true conditional data distribution $q(\mathrm{e}_0^{\mathrm{ta}} \mid \mathcal{T}_u^J, \{\mathcal{T}_u^1, \mathcal{T}_u^2, ..., \mathcal{T}_u^{J-1}\})$ in the embedding space should be estimated well, where $\mathbf{e}_0^{\mathrm{ta}}$ are embeddings of missing locations. Incorporating those conditions, based on the idea of the diffusion model, we need to learn the conditional transformation between

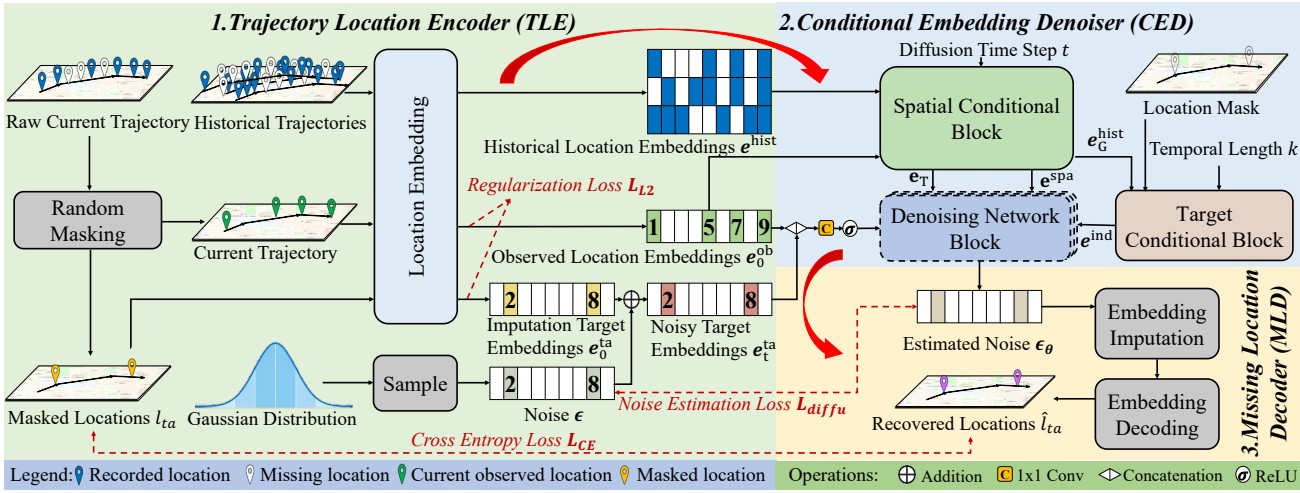

*Figure 1.* Overview of training stage of DiffMove.

consecutive steps (from $t$ to $t-1$):

$$p_\theta(\mathbf{e}_{t-1}^{\text{ta}} \mid \mathbf{e}_t^{\text{ta}}, \mathcal{T}_u^J, \{\mathcal{T}_u^1, \mathcal{T}_u^2, ..., \mathcal{T}_u^{J-1}\}) = \mathcal{N}(\mathbf{e}_{t-1}^{\text{ta}};$$
$$\boldsymbol{\mu}_\theta(\mathbf{e}_t^{\text{ta}}, t \mid \mathcal{T}_u^J, \{\mathcal{T}_u^1, \mathcal{T}_u^2, ..., \mathcal{T}_u^{J-1}\}), \quad (4)$$
$$\sigma_\theta(\mathbf{e}_t^{\text{ta}}, t \mid \mathcal{T}_u^J, \{\mathcal{T}_u^1, \mathcal{T}_u^2, ..., \mathcal{T}_u^{J-1}\})\mathbf{I})$$

Specifically, the parameterization of DDPM in Eq. (3) is also extended to the conditional case below similar to the proof in Tashiro et al. (2021):

$$\boldsymbol{\mu}_\theta(\mathbf{e}_t^{\text{ta}}, t \mid \mathcal{T}_u^J, \{\mathcal{T}_u^1, \mathcal{T}_u^2, ..., \mathcal{T}_u^{J-1}\}) =$$
$$\boldsymbol{\mu}^{\text{DDPM}}(\mathbf{e}_t^{\text{ta}}, t, \boldsymbol{\epsilon}_\theta(\mathbf{e}_t^{\text{ta}}, t \mid \mathcal{T}_u^J, \{\mathcal{T}_u^1, \mathcal{T}_u^2, ..., \mathcal{T}_u^{J-1}\}))),$$
$$\sigma_\theta(\mathbf{e}_t^{\text{ta}}, t \mid \mathcal{T}_u^J, \{\mathcal{T}_u^1, \mathcal{T}_u^2, ..., \mathcal{T}_u^{J-1}\}) = \sigma^{\text{DDPM}}(\mathbf{e}_t^{\text{ta}}, t)$$
$$(5)$$

where $\boldsymbol{\mu}_\theta(\boldsymbol{x}_t, t)$ and $\sigma_\theta(\boldsymbol{x}_t, t)$ in Eq. (3) are denoted as $\boldsymbol{\mu}^{\text{DDPM}}(\mathbf{e}_t, t, \boldsymbol{\epsilon}_\theta(\mathbf{e}_t, t))$ and $\sigma^{\text{DDPM}}(\mathbf{e}_t, t)$ here respectively, and general variable $\boldsymbol{x}$ is replaced by the embedding of missing location $\mathbf{e}$.

As can be observed, it essentially requires our denoising network $\boldsymbol{\epsilon}_\theta$ to incorporate observations in the current trajectory and historical trajectories. To well encode those conditions and realize the diffusion-based trajectory recovery, we present DiffMove, the training stage of which is shown in Figure 1. As the existing imputation work (Xia et al., 2021) did, DiffMove is trained in a self-supervised manner, which randomly masks some observed locations in the current trajectory and treats them as supervision signals, i.e., missing locations. To facilitate the description DiffMove, we decompose the whole process into three components: Trajectory Location Encoder (TLE), Conditional Embedding Denoiser (CED) and Missing Location Decoder (MLD).

### 4.1. Trajectory Location Encoder (TLE)

Trajectory Location Encoder (TLE) takes the current trajectory and historical trajectories of $J-1$ days as inputs,

and gives the embeddings of observed locations $e_0^{\text{ob}}$ and missing locations $e_t^{\text{ta}}$ in the current trajectory, and historical trajectories $e^{\text{hist}}$, which is shown in the left part of Figure 1. During the training stage, $e_t^{\text{ta}}$ is obtained by adding random Gaussian noise to embeddings of masked locations, while during the inference stage, $e_t^{\text{ta}}$ is directly sampled from the Gaussian distribution. We now elaborate on its training stage in detail as follows.

As shown in Figure 1, the blue and white location icons in trajectories represent observed and missing locations respectively. During the training stage, we first randomly mask some observed locations in the raw current trajectory as imputation targets, and thus separate it into the current trajectory (which is the actual input during the inference stage) and pseudo missing locations (icons in orange), i.e., masked locations $l_{ta}$. Locations in trajectories $l \in \mathcal{L}$ are represented by discrete IDs, and we also assign a special ID to the missing location, i.e., "null". After that, we feed the current trajectory, masked locations and historical trajectories into a location embedding layer, where each location $l$ would be transformed into a dense representation $\mathbf{e}^l \in \mathbb{R}^d$ by an embedding layer: $\mathbf{e}^l = \text{Embedding}_L(l)$ (note: other embedding methods are also welcome). The imputation target embeddings $e_0^{\text{ta}}$ are added with a Gaussian noise by Eq. (1) to form noisy target embeddings $e_t^{\text{ta}}$.

Finally, historical location embeddings $e^{\text{hist}}$, observed location embeddings $e_0^{\text{ob}}$ and noisy target embeddings $e_t^{\text{ta}}$ would be fed into Conditional Embedding Denoiser for the noise estimation. The learned embedding table, i.e., $\mathbf{E}_l \in \mathbb{R}^{(|\mathcal{L}|+1) \times d}$ would also be used to perform matching to recover missing locations, which would be introduced later in Section 4.3.

### 4.2. Conditional Embedding Denoiser (CED)

Conditional Embedding Denoiser (CED) takes diffusion time step $t$, noisy target embeddings $e_t^{\text{ta}}$ and conditions

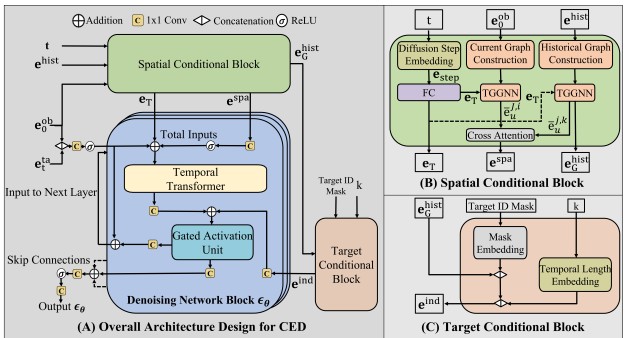

*Figure 2.* The Architecture for Conditional Embedding Denoiser

(i.e., historical location embeddings $e^{\text{hist}}$, observed location embeddings $e_0^{\text{ob}}$), and estimates the noise added to the target embeddings $e_{t-1}^{\text{ta}}$ at the time step, which is shown in the top right part of Figure 1. To fully exploit the power of conditions, Spatial Conditional Block is devised to model the transition and periodicity patterns, Target Conditional Block is designed to capture the relationship between the missing locations and historical trajectories, and Denoising Network Block is developed to capture the local context and produce the noise estimation. The design of each block is elaborated as follows.

**Spatial Conditional Block.** Spatial Conditional Block takes $e_0^{\text{ob}}$, $e^{\text{hist}}$ and diffusion step $t$, and gives the spatial condition $\mathbf{e}^{\text{spa}}$, which captures the transition and periodicity patterns from historical trajectories. In addition to $\mathbf{e}^{\text{spa}}$, an intermediate result, i.e., the diffusion time step embedding $\mathbf{e}_T$, is also passed to Denoising Network Block, and a historical trajectory embedding $\mathbf{e}_G^{\text{hist}}$ is also obtained to better capture the relationship between the missing locations and historical trajectories in Target Conditional Block.

Since graph neural network (GNN) has demonstrated its capability to capture the consecutive relationship between different entities (Xu et al., 2019; Wu et al., 2019) and attention mechanism is good at capturing the periodicity information (Liang et al., 2018), we propose to use GNN to learn the transition pattern and attention mechanism to learn the periodicity pattern. Since the degree of noise among different diffusion time steps is different, the importance of spatial conditions may also vary, we further incorporate the diffusion time step into the spatial condition learning. Considering above insights, we give the detailed structure of Spatial Conditional Block in Figure 2B. Firstly, we construct location transition graphs for both historical and current trajectories. For each trajectory, we construct an incoming and an outgoing transition graph, where all unique locations appearing in it serve as graph nodes, embeddings of locations from TLE serve as node embedding, and consecutive locations together form two adjacency matrices, i.e., $\mathbf{A}^I$ and $\mathbf{A}^O$, similar to (Xu et al., 2019).

Secondly, a Diffusion Step T Gated Graph Neural Network (TGGNN) is proposed to make diffusion-time-step-aware spatial pattern learning. Two TGGNN are introduced to learn patterns from current trajectory and historical trajectories, separately. We first transform the diffusion time step into a dense representation $\mathbf{e}_{\text{step}}$ by sinusoidal functions $\text{DiffEmbed}_T(t)$ (Kong et al., 2021; Tashiro et al., 2021), following by a fully connected layer: $\mathbf{e}_T = \text{DiffEmbed}_T(t)\mathbf{W}^T + \mathbf{b}^T$. Then, embeddings of current and historical trajectories would be passed into TGGNN for several times. In the s-th layer of TGGNN as shown in Figure 3, (1) the information propagation from neighborhood is performed based on node embeddings of s-th layer $[\mathbf{e}_s^1, \ldots, \mathbf{e}_s^N]$ and two adjacency matrices, i.e., $\mathbf{A}^I$ and $\mathbf{A}^O$ to obtain incoming/outgoing aggregated node embedding $\mathbf{e}_{I,s}/\mathbf{e}_{O,s}$, respectively; (2) an intermediate representations $\mathbf{a}_{s+1}$ is created by concatenating those aggregated node embeddings with embedded diffusion time step $\mathbf{e}_T$ to enhance the representations; (3) a gating mechanism (Li et al., 2016) is used to fuse the node embeddings of the s-th layer and (s+1)-th layer:

$$\mathbf{e}_{I,s} = \left(\mathbf{A}_i^I \left([\mathbf{e}_s^1, \ldots, \mathbf{e}_s^N]\mathbf{W}^I + \mathbf{b}^I\right)\right)$$
$$\mathbf{e}_{O,s} = \left(\mathbf{A}_i^O \left([\mathbf{e}_s^1, \ldots, \mathbf{e}_s^N]\mathbf{W}^O + \mathbf{b}^O\right)\right) \quad (6)$$
$$\mathbf{a}_{s+1} = \mathbf{e}_{I,s} \| \mathbf{e}_{O,s} \| \mathbf{e}_T, \quad \mathbf{e}_{s+1}^l = \text{Gates}(\mathbf{a}_{s+1}, \mathbf{e}_s)$$

where $\|$ is the concatenation, $\mathbf{W}^I, \mathbf{W}^O, \mathbf{W}^T \in \mathbb{R}^{d \times d}$ are learnable parameters, and $\mathbf{b}^I, \mathbf{b}^O, \mathbf{b}^T \in \mathbb{R}^d$ are bias vectors, $N$ is the number of unique locations in the trajectory, and Gates denotes several gates (Li et al., 2016), i.e., update gate, reset gate, to fuse node embeddings in consecutive layers. We denote the final node embedding of TGGNN as $\overline{\mathbf{e}}^l$ for simplicity. By pooling final node embeddings from the historical branch, the historical trajectory embedding $\mathbf{e}_G^{\text{hist}}$ is derived.

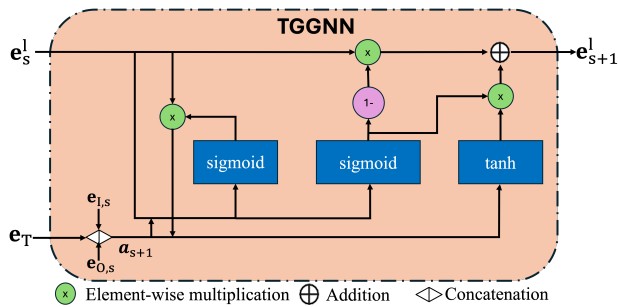

*Figure 3.* Design of TGGNN.

Thirdly, we employ CrossAttention (Xia et al., 2021) to capture the periodicity among the current trajectory and historical trajectories. For each user $u$, the $\text{head}_h$ calculate

the cross attention between the $i$-th time slot of the current trajectory embedding ($\overline{\mathbf{e}}_u^{J,i}$) and the $k$-th time slot of the $j$-th historical trajectory embedding ($\overline{\mathbf{e}}_u^{j,k}$). The final spatial condition $\mathbf{e}^{\text{spa}}$ is generated by a linear projection of the concatenation of $H$ number of heads as shown in Eq. (7).

$$\text{head}_h = \text{CrossAttention}(\overline{\mathbf{e}}_u^{J,i}, \overline{\mathbf{e}}_u^{j,k})$$
$$\mathbf{e}^{\text{spa}} = ReLU(\mathbf{W}(\text{head}_1 \| \ldots \| \text{head}_H) + \overline{e}_u^{J,i}) \quad (7)$$

**Target Conditional Block.** Target Conditional Block takes target ID mask, temporal length $k$ and historical trajectory embedding $\mathbf{e}_G^{\text{hist}}$, and gives $\mathbf{e}^{\text{ind}}$, which captures the correlations between the missing locations and historical trajectories to help the inference. The target ID mask representations are in one-hot form: non-target locations are represented with all zeros, while target locations are represented with ones. This method creates a placeholder embedding that signifies the absence of data at specific positions. This embedding is then concatenated with $\mathbf{e}_G^{\text{hist}}$ and will be fused with the output of the Temporal Length Embedding layer. We incorporate temporal length embedding $\mathbf{k} = \{k_{1:K}\}$ as auxiliary information. We adopt a 128-dimensional temporal embedding, consistent with prior research (Vaswani et al., 2017; Zuo et al., 2020):

$$\mathbf{k}_{\text{embedding}}(k_l) = \Big(\sin(k_l/\tau^{0/64}), \ldots, \sin(k_l/\tau^{63/64}),$$
$$\cos(k_l/\tau^{0/64}), \ldots, \cos(k_l/\tau^{63/64})\Big) \quad (8)$$

Here, $\tau = 10000$. This temporal length embedding enriches the model with sequential information, enhancing its ability to recover trajectories. Finally, the concatenated representation forms $\mathbf{e}^{\text{ind}}$.

**Denoising Network Block.** This block is the function of DiffMove to model $\epsilon_\theta$ in Eq. (5). It receives the inputs from TLE together with $\mathbf{e}_T$, $\mathbf{e}^{\text{spa}}$, and $\mathbf{e}^{\text{ind}}$. The concatenation of $\mathbf{e}_0^{\text{ob}}$ and $\mathbf{e}_t^{\text{ta}}$ from TLE as well as $\mathbf{e}^{\text{spa}}$ are passed through a 1D convolution layer and a ReLU, and the results of both would be added to $\mathbf{e}_T$ to form the the total input of a temporal transformer (Tashiro et al., 2021) (with multihead self-attentions) to learn the temporal sequence features. Then it will be passed through a 1D convolution layer and added with the 1D convoluted result of $\mathbf{e}^{\text{ind}}$. Following a gated activation unit (Ramachandran et al., 2017), part of outputs is directed to the next residual layer as input, whereas the remainder is incorporated into the final output via a skip connection. The Conv1 $\times$ 1 blocks in the network facilitate the mapping of data to suitable dimensions. Ultimately, the output $\hat{\epsilon}$ is the culmination of data passed through skip-connections from each residual layer.

### 4.3. Missing Location Decoder (MLD)
Missing Location Decoder (MLD) leverages CED to recover locations. It consists of two steps: Embedding Im-

putation, which transforms noises into meaningful location embeddings based on the estimated noise from CED, and Embedding Decoding, which decodes the estimated target embeddings to locations to recover the trajectory.

**Embedding Imputation.** Embedding Imputation is to obtain robust estimated target embeddings, we perform target embeddings generation for $M$ times, and the means of target embeddings $\bar{\hat{\mathbf{e}}}_0^{\text{ta}}$ are used for the location decoding. For each time of generation, a random noise $\mathbf{e}_T^{\text{ta}}$ is sampled from $\mathcal{N}(\mathbf{0}, \mathbf{I})$, then we perform the reverse process of diffusion from step $T$ to 1 gradually according to Eq. (4) and Eq. (5) to obtain one estimated target embedding $\hat{\mathbf{e}}_0^{\text{ta}}$.

**Embedding Decoding.** After the mean imputed target embeddings $\bar{\hat{\mathbf{e}}}_0^{\text{ta}}$ are obtained, we calculate the inner product between $\bar{\hat{\mathbf{e}}}_0^{\text{ta}}$ and location embeddings in embedding table $\mathbf{E}_l' \in \mathbb{R}^{|L|}$, which is from $\mathbf{E}_l$ in TLE after excluding the embedding of "null" item. For each imputed target embedding $\bar{\hat{\mathbf{e}}}_0^{ta,i}$, its similarities to different locations $\hat{\mathbf{z}}_i^{\text{ta}} \in \mathbb{R}^{|\mathcal{L}|}$ are calculated as follows: $\hat{\mathbf{z}}_i^{\text{ta}} = \bar{\hat{\mathbf{e}}}_0^{ta,i} \mathbf{E}_l'^\top$

Subsequently, we apply a softmax function to obtain the location likelihood vector $\hat{\mathbf{y}}^i$ for each imputation target: $\hat{\mathbf{y}}^i = \text{softmax}(\hat{\mathbf{z}}_i^{\text{ta}})$. During the inference stage, the location with the highest probability would be used to recover the trajectory.

### 4.4. Model Training
Since trajectory recovery results are discrete, which cannot be easily obtained by the denoising network, multiple losses are introduced as shown in Figure 1 when we train DiffMove.

The first loss is diffusion loss $L_{\text{diffu}}$, which calculates noise estimation accuracy. We sample a noise $\epsilon$ and obtain the noisy target embeddings $\mathbf{e}_t^{\text{ta}}$ at the diffusion time step $t$ by Eq. (1). Then, DiffMove estimates the added noise conditioned on observed locations in the current trajectory and the historical trajectories. The expectation of the mean squared error between the actual noise and the estimated noise is served as $L_{\text{diffu}}$, which is defined as follows:

$$L_{\text{diffu}}(\theta) = \mathbb{E}_{\mathbf{e}_0^{\text{ta}} \sim q(\mathbf{e}_0^{\text{ta}}), \epsilon \sim \mathcal{N}(\mathbf{0}, \mathbf{I}), t} \| (\epsilon -$$
$$\epsilon_\theta(\mathbf{e}_t^{\text{ta}}, t \mid \mathcal{T}_u^J, \{\mathcal{T}_u^1, \mathcal{T}_u^2, ..., \mathcal{T}_u^{J-1}\})) \|_2^2 \quad (9)$$

The second loss characterizes the location recovery accuracy, which is the cross entropy loss. Given the one-hot representations of masked locations in the raw current trajectory $\mathbf{Y} = \{\mathbf{y}^1, \mathbf{y}^2, ..., \mathbf{y}^{K_{ta}}\}$ and the predicted likelihood of the imputed locations $\hat{\mathbf{Y}}^j = \{\hat{\mathbf{y}}^1, \hat{\mathbf{y}}^2, ..., \hat{\mathbf{y}}^{K_{ta}}\}$ ($K_{ta}$ is the number of masked locations in the raw current trajectory), $L_{\text{CE}}$ is defined as

$$L_{\text{CE}}(\mathbf{Y}, \hat{\mathbf{Y}}) = -\sum_{j=1}^{K_{ta}} \sum_{i=1}^{|\mathcal{L}|} y_i^j \log(\hat{y}_i^j) \quad (10)$$

The third loss is an L2 loss for regularization, which is suggested in (Gong et al., 2022). It regularizes the learning of the location embeddings of the raw current trajectory, i.e.,

$$L_{\text{L2}}(\mathbf{e}_0^{\text{ta}}, \mathbf{e}_0^{\text{ob}}) = \frac{1}{Kd}(\sum_{i=1}^{K_{ta}} ||\mathbf{e}_0^{ta,i}||^2 + \sum_{j=1}^{K-K_{ta}} ||\mathbf{e}_0^{ob,j}||)^2 \tag{11}$$

Consequently, DiffMove is trained end to end by jointly optimizing the above three types of losses: $L_{\text{E2E}} = L_{\text{diffu}} + \lambda_1 L_{\text{CE}} + \lambda_2 L_{\text{L2}}$, where $\lambda_1$ and $\lambda_2$ are multi-task learning weights.

## 5. Experiments

### 5.1. Datasets

- **Foursquare**[2]: This dataset (Yang et al., 2014) was obtained from the Foursquare API, covering the period from April 2012 to February 2013. Each record in the dataset includes user ID, timestamp, GPS location, and POI ID. We standardize the timestamps to a one-week format while preserving the original trajectory order.

- **Geolife**[3]: This publicly available dataset is sourced from the Microsoft Research Asia Geolife project (Zheng et al., 2010), involving 182 users and spanning from April 2007 to August 2012 globally. Each trajectory is represented by a sequence of time-stamped points, providing longitude and altitude information (Zheng et al., 2010).

### 5.2. Baselines

We evaluate the proposed approach against baseline methods, including both traditional approaches grounded in our understanding of human mobility and advanced deep learning models capable of capturing intricate mobility patterns. We evaluate the proposed approach against below baselines: **Rule-based methods**: 1): Top, 2) Markov (Gambs et al., 2012), 3) PMF (Mnih & Salakhutdinov, 2007). **Deep learning based methods**: 4) LSTM (Liu et al., 2016), 5) BiL-STM (Zhao et al., 2018), 6) DeepMove (Feng et al., 2018), 7) AttnMove (Xia et al., 2021), 8) PeriodicMove (Sun et al., 2021), 9) TRILL (Deng et al., 2023). Selections of baselines are to ensure fair comparisons in the same setting of free-space human trajectory recovery. More details about baselines will be provided in the Appendices.

### 5.3. Experimental Settings

Following (Deng et al., 2023), we mask randomly 10 time slots per day for both the Geolife and Foursquare dataset. The trajectories are split chronologically into training (60%),

---

[2] https://sites.google.com/site/yangdingqi/home/foursquare-dataset/

[3] https://www.microsoft.com/en-us/research/project/geolife-building-social-networks-using-human-location-history/

validation (20%) and test (20%) sets. We utilize the widely adopted metrics Recall@K and Mean Average Precision (MAP) (Wang et al., 2019). Recall@K measures whether the ground truth is present in the top K predictions, averaged over all test cases. MAP evaluates the overall ranking quality by considering the entire prediction list. Larger values for both metrics indicate better performance. Additionally, we use Distance@K, which computes the smallest geographical distance between the centers of locations in the top-K ranked list and the ground truth, averaged across test cases. Lower Distance@K signifies better performance. We report experimental results for Recall@K and Distance@K at K = 1, 5 and 10. This allows a comprehensive assessment of our model's ability to rank ground truth locations.

### 5.4. Experiment Results

As shown in Table 1, firstly, rule-based methods fail to achieve high accuracy, exhibiting the worst performance for all evaluation metrics on both datasets. Secondly, RNN-based methods perform better than rule-based methods as they can model simple sequential patterns among locations. Bidirectional RNNs perform better than unidirectional ones, indicating the importance of spatial-temporal constraints for human mobility recovery. State-of-the-art deep learning methods, including AttnMove, PeriodicMove and TRILL achieve satisfactory performance by capturing sequential patterns and simple periodicity of human mobility. However, DiffMove outperforms all the baselines for all evaluation metrics on both datasets. Specifically, for Recall, DiffMove outperforms the best baseline, TRILL, by 9.57% on Geolife dataset and by 11.56% on Foursquare dataset. For Distance, DiffMove outperforms the best baseline, TRILL, by 19.66% on Geolife dataset and by 7.93% on Foursquare dataset. For MAP, DiffMove outperforms the best baseline, TRILL, by 7.05% on Geolife dataset and by 9.56% on Foursquare dataset. These significant improvements indicate that our proposed DiffMove can better learn spatial temporal patterns of both current and historical trajectories and recover the details of human mobility. We also change the number of generated samples M (in Section 4.3) from 4 to 1, which simulates the normal single prediction method. We observe the reduced performances in Table 1 (-1.85% Recall@1 on Geolife and -1.7% Recall@1 on Foursquare) due to lacking probabilistic generation and sampling, which highlights the significance of the probabilistic generation instead of deterministic single imputed embedding.

### 5.5. Ablation Analysis

We conduct ablations by systematically removing individual components. The results of Foursquare dataset are presented in Table 2. The recall, distance and MAP performance of the first ablation with unconditional diffusion (No observed location, no spatial, and target condition involved) drops significantly to almost nonfunctional status. This emphasizes

*Table 1.* Overall performance comparison in terms of Recall@K, Distance@K, and MAP.

| Dataset | Methods | Recall@K | | | Distance@K | | | MAP |
|---------|---------|----------|----------|-----------|------------|------------|-------------|------|
| | | Recall@1 | Recall@5 | Recall@10 | Distance@1 | Distance@5 | Distance@10 | |
| Geolife | Top | 0.1148 | 0.2451 | 0.3166 | 7863 | 6259 | 5176 | 0.1812 |
| | Markov | 0.1417 | 0.3263 | 0.3974 | 6909 | 4974 | 4259 | 0.2304 |
| | PMF | 0.1941 | 0.3436 | 0.4059 | 6506 | 4389 | 3555 | 0.2752 |
| | LSTM | 0.2086 | 0.3917 | 0.4720 | 6318 | 3928 | 3068 | 0.2965 |
| | BiLSTM | 0.2285 | 0.4538 | 0.5773 | 6209 | 3620 | 2255 | 0.3298 |
| | DeepMove | 0.3045 | 0.5380 | 0.6371 | 5370 | 2052 | 1358 | 0.4131 |
| | AttnMove | 0.3920 | 0.6696 | 0.7213 | 5342 | 2007 | 975 | 0.5046 |
| | PeriodicMove | 0.4199 | 0.6893 | 0.7681 | 4209 | 1443 | 863 | 0.5385 |
| | TRILL | 0.4721 | 0.7563 | 0.8364 | 3484 | 1112 | 603 | 0.5985 |
| | DiffMove w/ single gen-sample | 0.4988 | 0.7701 | 0.8350 | 2905 | 973 | 601 | 0.6180 |
| | **DiffMove** | **0.5173** | **0.7987** | **0.8578** | **2799** | **708** | **444** | **0.6407** |
| | **%Improvement** | **9.57%** | **5.61%** | **2.56%** | **19.66%** | **36.33%** | **26.37%** | **7.05%** |
| Foursquare | Top | 0.0865 | 0.1673 | 0.2268 | 8427 | 4919 | 3483 | 0.1347 |
| | Markov | 0.1090 | 0.2010 | 0.2575 | 8345 | 4402 | 3125 | 0.1792 |
| | PMF | 0.1215 | 0.2468 | 0.2887 | 8116 | 3971 | 3229 | 0.2358 |
| | LSTM | 0.1393 | 0.2540 | 0.3143 | 7913 | 3804 | 2801 | 0.2519 |
| | BiLSTM | 0.2323 | 0.3968 | 0.4703 | 6206 | 2745 | 1849 | 0.3154 |
| | DeepMove | 0.2612 | 0.4631 | 0.5337 | 5189 | 2648 | 1649 | 0.3789 |
| | AttnMove | 0.2975 | 0.5172 | 0.5746 | 4942 | 2396 | 1482 | 0.4078 |
| | PeriodicMove | 0.3125 | 0.5534 | 0.6264 | 4704 | 1758 | 1197 | 0.4245 |
| | TRILL | 0.3227 | 0.5636 | 0.6372 | 4639 | 1650 | 1074 | 0.4341 |
| | DiffMove w/ single gen-sample | 0.3430 | 0.4614 | 0.5009 | 5206 | 1964 | 1339 | 0.4035 |
| | **DiffMove** | **0.3600** | **0.6090** | **0.6876** | **4271** | **1548** | **989** | **0.4756** |
| | **%Improvement** | **11.56%** | **8.06%** | **7.91%** | **7.93%** | **6.18%** | **7.91%** | **9.56%** |

the inadequacy of relying solely on the default diffusion probabilistic model for the trajectory recovery task in latent space and underscores the importance of integrating multiple spatial and temporal related specific conditional modules for effective learning and training. The removal of the Temporal Transformer or Spatial Conditional Block significantly impacts performance, emphasizing their critical roles in reinforcing spatial and temporal constraints for missing locations, resulting in substantial improvement when leveraging historical information. The removal of the Target Conditional Block leads to decreased model performance, highlighting the role of the target condition in guiding the model to reconstruct specific embeddings in the locations through the diffusion process. Additionally, the Missing Location Decoder is also identified as a crucial component. It can not be compared in the table since its removal renders the model nonfunctional, as this module plays a vital role in converting the reconstructed embeddings of missing locations into a decoded discrete ID space.

*Table 2.* Impact of components on Foursquare dataset, where $\delta$ denoted the performance decline.

| Ablation | Recall($\Delta$) | Dis.($\Delta$) | MAP($\Delta$)(m) |
|----------|------------------|----------------|------------------|
| Unconditional | 0.0416 (-88.44%) | 7913 (-85.27%) | 0.0944 (-80.15%) |
| Spatial Conditional Block | 0.3382 (-6.06%) | 4591 (-7.49%) | 0.4496 (-5.47%) |
| Target Conditional Block | 0.3493 (-2.97%) | 4377 (-2.48%) | 0.4632 (-2.61%) |
| Temporal Transformer | 0.3023 (-16.03%) | 4783 (-11.99%) | 0.4166 (-12.41%) |

### 5.6. Robustness Study

As shown in Table 3, our proposed model, DiffMove, consistently outperforms the baseline models, AttnMove, Periodic-Move and TRILL across various missing ratios. The second best results are underlined and the improvements are listed in the brackets. Notably, as the percentage of missing locations in historical trajectories increases from 20% to 80%, DiffMove exhibits superior performance, achieving higher Recall@10, lower Dist@10, and improved MAP scores compared to the baselines. This suggests that DiffMove is more robust in scenarios with higher missing percentages of historical trajectories and sparser locations. The significant reduction in Dist@10 for DiffMove indicates its effectiveness in accurately recovering missing locations. Remarkably, the Distance metric performance of our DiffMove with 80% missing ratio even outperforms TRILL with 40% missing rate and surpasses both PeriodicMove and Attn-move, even when they have lower missing rates 20%. This serves as one aspect of scalability and further reinforces the efficacy and good potential of DiffMove in handling larger datasets since it shows better performance even when the model is utilizing a smaller portion of the same existing data (larger missing ratio than those of baselines), which provides insights into its applicability across various scalability of missing ratio scenarios. These results underscore the robust, scalable and superior performance of DiffMove in more challenging task of trajectory recovery, making it a

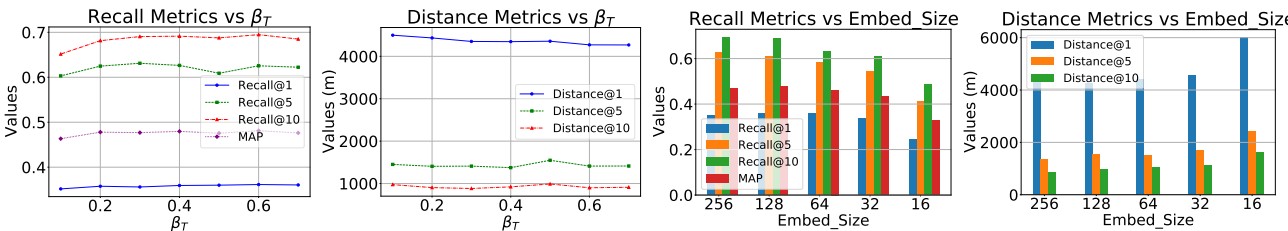

*Figure 4.* Recall vs $\beta_T$    *Figure 5.* Distance vs $\beta_T$    *Figure 6.* Recall vs Embed_Size    *Figure 7.* Distance vs Embed_Size

promising model for real-world applications.

*Table 3.* Performance w.r.t. Missing Ratios on Geolife

| Methods | Metrics | Missing Rate | | | |
|---|---|---|---|---|---|
| | | 20% | 40% | 60% | 80% |
| **AttnMove** | Recall@10 | 0.7117 | 0.6985 | 0.6785 | 0.6160 |
| | Dist@10 | 987 | 1037 | 1174 | 1371 |
| | MAP | 0.4815 | 0.4657 | 0.4226 | 0.4112 |
| **PeriodicMove** | Recall@10 | 0.7451 | 0.7392 | 0.7186 | 0.6857 |
| | Dist@10 | 884 | 954 | 1059 | 1176 |
| | MAP | 0.5175 | 0.4750 | 0.4413 | 0.4076 |
| **TRILL** | Recall@10 | 0.8216 | 0.8038 | 0.7627 | 0.7436 |
| | Dist@10 | 682 | 720 | 915 | 1089 |
| | MAP | 0.5760 | 0.5534 | 0.5111 | 0.5044 |
| **DiffMove** | Recall@10 | **0.8344** | **0.8163** | **0.7931** | **0.7863** |
| | Dist@10 | **507** | **617** | **681** | **695** |
| | MAP | **0.6107** | **0.5730** | **0.5495** | **0.5099** |

## 5.7. Parameter Study

We also conduct some experiments to provide insights into the performance of our model (DiffMove) across different values of $\beta_T$ and embedding size.

*Beta_end $\beta_T$:* Figure 4 and 5 illustrate the interplay between Recall@K, MAP, and Distance@K across different values of $\beta_T$. We vary the $\beta_T$ to change the noise schedule, the Recall@1 and Distance@1 performance are more important and seem to have increasing trends but drop when $\beta_T$ is too large although there are some fluctuations for Recall@5 and Recall@10. We try to choose the optimal value at 0.6 after consideration of all tradeoffs. The relationship between $\beta_T$ and spatial Distance@1 accuracy reveals specific $\beta_T$ values that result in optimal spatial alignment, indicating the importance of $\beta_T$ in shaping spatial aspects of trajectory recovery performance.

*Embedding Size:* In addition to $\beta_T$, Figure 6 and 7 illustrate the variation in all metrics across different embedding sizes. As expected, the initial increase of the embedding size contributes to the increase of Recall@1 and Distance@1 since more information is recorded by embedded vectors. However, too large embedding could also bring some uncertain information and lead to saturation of prediction accuracy. As a result, we choose the optimal value at 128.

## 6. Conclusion

In conclusion, this research addresses the problem of trajectory recovery from sparse human mobility data by introduc-

ing a novel model, DiffMove. Leveraging an embedding-space conditional diffusion framework, it excels in trajectory recovery by constructing and utilizing conditional information with trajectory spatial patterns, inter-trajectory dependencies, temporal and target location patterns. The model is innovatively designed and integrated with multiple conditional feature extraction modules, tackling the complexity of spatial temporal dependencies in a principled manner. Our extensive experiments demonstrate that DiffMove outperforms all state-of-the-art baselines, showcasing its effectiveness in recovering missing locations. While DiffMove demonstrates strong performance under random masking, real-world uncertainty may involve structured missingness (e.g., persistent gaps due to device failures). Future work could explore models conditioned on specific missing patterns.

## Acknowledgements

This work was supported by the National Key R&D Program of China (No. 2023YFC2308703) and the National Natural Science Foundation of China (No. 62306033). This research is also supported in part by Alibaba Group through Alibaba-NTU Singapore Joint Research Institute (JRI), Nanyang Technological University, Singapore, and by the Ministry of Education, Singapore, under its Academic Research Fund (Tier 2 Award MOE-T2EP20221-0013 and Tier 1 Award (RG20/24)). Any opinions, findings and conclusions or recommendations expressed in this material are those of the author(s) and do not reflect the views of the Ministry of Education, Singapore.

## Impact Statement

This paper presents work whose goal is to advance the field of Machine Learning. There are potential societal consequences of our work, none which we feel must be specifically highlighted here.

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

# A. Appendix / Supplemental material

## A.1. More Details on Transition Graph Construction of Spatial Conditional Block

The prior investigations (Xu et al., 2019; Wu et al., 2019; Sun et al., 2021) have demonstrated that a gated graph neural network (GGNN) is adept at capturing intricate transition patterns among nodes. This characteristic renders the gated GNN well-suited for addressing our specific problem. In the graph neural network layer, we handle each trajectory independently to unveil the complex transition patterns concealed within each trajectory. To elaborate, we initiate the process by establishing a directed graph for each trajectory. Subsequently, the gated GNN is employed on each of these directed graphs to refine the location embeddings, thereby capturing the transition patterns into the model. Besides this, we manage to conduct data fusion of diffusion step embedding into the gated GNN to make the transition pattern learning adaptive to the diffusion time step.

**Trajectory Graph Construction:**   The initial step of the graph neural network layer involves constructing a transition graph representation for each historical and current trajectory in the context of trajectory recovery. Similar to session recommendation, given a location IDs' trajectory $\mathcal{T} : l_1 \to l_2 \ldots \to l_K$, we consider each location $l_i$ as a node and $(l_{i-1}, l_i)$ as an edge, representing the user's movement from $l_{i-1}$ to $l_i$ in the trajectory $\mathcal{T}$. Consequently, each trajectory can be conceptualized as a directed graph. The graph structure is learned by facilitating communication among distinct nodes. Specifically, let $A^I$, $A^O$ denote the weighted transitions of incoming and outgoing edges in the trajectory graph, respectively. To address the possibility of repeated occurrences of locations in a trajectory, we assign each edge a normalized weight, calculated as the edge's occurrence divided by the outdegree of the start node of that edge. Consider transitions in a trajectory $[l_1, l_2, l_3, l_2, l_4]$, the corresponding graph, the incoming matrix $\mathbf{A}^I$ and the outgoing matrix $\mathbf{A}^O$ are shown in Figure 8.

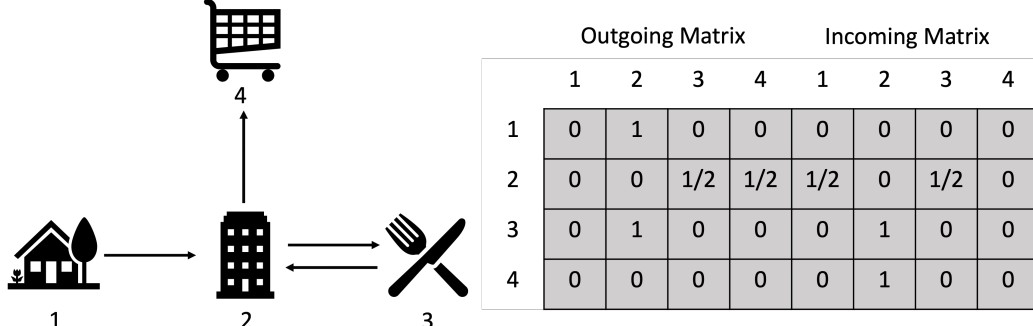

*Figure 8.* A example of a trajectory transition graph and the incoming and outgoing matrix **A**

## A.2. Details on Diffusion Step Embedding

$\mathbf{e}_{\text{step}}$ is the 128-dimension diffusion step embedding obtained from a special embedding layer $\text{DiffEmbed}_T(t)$ by sinusoidal functions following previous works (Kong et al., 2021; Tashiro et al., 2021):

$$
\text{DiffEmbed}_T(t) = \Big( \sin(10^{0 \cdot 4/63} t), \ldots,
$$
$$
\sin(10^{63 \cdot 4/63} t), \cos(10^{0 \cdot 4/63} t), \ldots, \cos(10^{63 \cdot 4/63} t) \Big) \tag{12}
$$

and it is further processed through a fully connected layer to obtain $\mathbf{e}_T$.

## A.3. Details of Cross Attention in Spatial Conditional Block

Further details of Eq. (7) are elucidated in Eq. (13).

$$\alpha_{i,k}^{(h)} = \frac{\exp(\phi^{(h)}(\overline{e}_u^{J,i}, \overline{e}_u^{j,k}))}{\sum_{g=1}^{K} \exp(\phi^{(h)}(\overline{e}_u^{J,i}, \overline{e}_u^{j,g}))},$$

$$\phi^{(h)}(\overline{e}_u^{J,i}, \overline{e}_u^{j,k}) = \langle W_Q^{(h)} \overline{e}_u^{J,i}, W_K^{(h)} \overline{e}_u^{j,k} \rangle,$$

$$\widetilde{e}_u^{j,i(h)} = \sum_{k=1}^{K} \alpha_{i,k}^{(h)} (W_V^{(h)} \overline{e}_u^{j,k}),$$  (13)

$$\widetilde{e}_u^{j,i} = \widetilde{e}_u^{j,i(1)} \parallel \widetilde{e}_u^{j,i(2)} \parallel \cdots \parallel \widetilde{e}_u^{j,i(H)},$$

$$\mathbf{e}^{\text{spa}} = ReLU(W\widetilde{e}_u^{j,i} + \overline{e}_u^{J,i}),$$

where $W_Q^{(h)}, W_K^{(h)}, W_V^{(h)} \in \mathbb{R}^{d' \times d}$ are transformation matrices, and $\langle , \rangle$ denotes the inner product function. Next, we compute the representation of time slot $i$ for each head by aggregating information from all locations in other time slots based on the coefficients $\alpha_{i,k}^{(h)}$. The symbol $\parallel$ denotes the concatenation operator, and $H$ represents the total number of heads.

### A.4. Imputation (Sampling) Algorithm with DiffMove

---
**Algorithm 1** Imputation (Sampling) with DiffMove

---
1: **Input:** a Location Embedding sample $\mathbf{e}_0$, No. of generated samples $M$, trained denoising function $\epsilon_\theta$
2: **Output:** Imputed missing value $\bar{\hat{\mathbf{e}}}_0^{\text{ta}}$
3: Construct observation condition of $\mathbf{e}_0$ as $\mathbf{e}_0^{\text{ob}}$
4: **for** $m = 1$ **to** $M$ **do**
5:    $\mathbf{e}_T^{\text{ta}} \sim \mathcal{N}(\mathbf{0}, \mathbf{I})$ where the dimension of $\mathbf{e}_T^{\text{ta}}$ corresponds to the missing indices of $\mathbf{e}_0$
6:    **for** $t = T$ **to** 1 **do**
7:       Sample $\hat{\mathbf{e}}_{t-1}^{\text{ta}}$ using Eq. (4) and Eq. (5)
8:    **end for**
9:    Record $\hat{\mathbf{e}}_0^{\text{ta}}$
10: **end for**
11: Calculate mean value $\bar{\hat{\mathbf{e}}}_0^{\text{ta}}$ by $mean(\hat{\mathbf{e}}_0^{\text{ta}})$

---

### A.5. Study of Number of Generated Samples M

We change the number of generated samples M from 4 to 1 (meaning only predict one single embedding and use it directly), which simulates the normal deterministic way as an ablation study to some extent. We observe the Table 4 results of reduced performances of the original DiffMove (-1.85% Recall@1 on Geolife and -1.7% Recall@1 on Foursquare) due to lack of considering effects of probabilistic generation and sampling , which highlights the significance of the probabilistic generation instead of deterministic single imputed embedding.

### A.6. Implementation Details

DiffMove is trained using batch gradient descent with the Adam optimizer (Kingma & Ba, 2014), implemented in Python and PyTorch (Paszke et al., 2019), on a Linux server equipped with an NVIDIA RTX A5000. We set random seed as 2021. Multi-task learning weights $\lambda_1$ and $\lambda_2$ are set as 1 after experimental study. We employed a learning rate of 0.001 with a weight decay of 1e-6. We set the location embedding size as 128, the steps (loops) of TGGNN as 2, the number of heads for cross attention as 4, diffusion step embedding dimension and temporal length embedding dimension are 128.

| Dataset | Methods | Recall@1 | Recall@5 | Recall@10 | Distance@1 | Distance@5 | Distance@10 | MAP |
|---------|---------|----------|----------|-----------|------------|------------|-------------|-----|
| Geolife | DiffMove w/ single gen-sample | 0.4988 | 0.7701 | 0.8350 | 2905 | 973 | 601 | 0.6180 |
| | DiffMove | 0.5173 | 0.7987 | 0.8578 | 2799 | 708 | 444 | 0.6407 |
| Foursquare | DiffMove w/ single gen-sample | 0.3430 | 0.4614 | 0.5009 | 5206 | 1964 | 1339 | 0.4035 |
| | DiffMove | 0.3600 | 0.6090 | 0.6876 | 4271 | 1548 | 989 | 0.4756 |

*Table 4.* Performance comparison between DiffMove variants

We set the number of residual layers as 4, residual channels as 128, and attention heads for the temporal transformer as 8. We set the number of the diffusion step $T = 50$, the minimum noise level $\beta_1 = 0.0001$, and the maximum noise level $\beta_T = 0.6$. We tuned hyperparameters for each dataset to achieve optimal results. Following recent studies (Song et al., 2021; Nichol & Dhariwal, 2021), a quadratic schedule was adopted for decay of $\alpha_t$ to enhance sample quality: $\beta_t = \left( \frac{T-t}{T-1} \sqrt{\beta_1} + \frac{t-1}{T-1} \sqrt{\beta_T} \right)^2$.

### A.7. Details of Baselines

We evaluate the proposed approach against several baseline methods, including both traditional approaches grounded in our understanding of human mobility and advanced deep learning models capable of capturing intricate mobility patterns:

- Top: A straightforward counting-based method that selects the most frequently visited location in the training set as the recovery for each user.

- Markov (Gambs et al., 2012): A commonly used method treating visited locations as states and constructing a transition matrix to capture first-order transition probabilities.

- PMF (Mnih & Salakhutdinov, 2007): An advanced model rooted in conventional collaborative filtering, based on the user location matrix.

- LSTM (Liu et al., 2016): A deep learning model that captures sequential patterns through recurrent neural networks, using the predicted next time slot as recovery.

- BiLSTM (Zhao et al., 2018): An extension of LSTM with bidirectional recurrent neural networks, incorporating spatial-temporal constraints from all observed locations.

- DeepMove (Feng et al., 2018): A model that jointly considers user preferences and sequential dependencies for predicting the next location used for recovery.

- AttnMove (Xia et al., 2021): A method leveraging various attention mechanisms to capture the regularity and patterns in a user's mobility.

- PeriodicMove (Sun et al., 2021):A recent model that considers factors such as transition patterns among locations and periodicity in human mobility.

- TRILL (Deng et al., 2023): The latest state-of-the-art model which is capturing global mobility patterns leveraging graph convolutional networks for mobility patterns.

*Table 5.* Basic statistics of mobility datasets.

| Dataset | City | #Historical Trajs | #Current Trajs | #Distinctive Locations | Total #IDs Processed in Training (approx.) |
|---------|------|-------------------|----------------|------------------------|---------------------------------------------|
| Foursquare | Tokyo | 11,430 | 2,286 | 1,411 | 404.9K |
| Geolife | Beijing | 15,648 | 3,912 | 1,124 | 563.3K |

### A.8. Pre-processing

For our location representation, We collect the cities' street map data of Tokyo and Beijing from an online map source and partition the region into distinct blocks. Each of these blocks is considered as an individual location, with an average area size of approximately 0.25 $km^2$ (500m x 500m for both datasets). Other pre-processings are the same as (Deng et al., 2023). As per (Chen et al., 2019), a 30-minute time interval is employed for both datasets. Further details and statistics are presented in Table 5. Our framework is not inherently tied to the 30-minute interval and can easily adapt to other interval lengths, such as 10 minutes or even finer resolutions, provided the data supports such granularity. We can always treat the time interval as a parameter to tune on the data side. This flexibility makes our method broadly applicable, as it caters to human mobility patterns by modeling discrete location transitions effectively. Human mobility typically involves meaningful transitions at specific timeframes (e.g., work, shopping, dining), which align naturally with discrete intervals.

## A.9. Scalability Study

Based on the experimental results shown in Figures 9 and 10, we conducted a scalability study comparing the performance of the DiffMove and best baseline model TRILL across different scaling ratios ranging from 20% to 100% to vary the scale of the whole training dataset. Regarding recall performance, as depicted in Figure 9, DiffMove consistently outperforms TRILL across all scaling ratios. In terms of distance performance, illustrated in Figure 10, both models display an increasing trend with higher scaling ratios. Notably, DiffMove maintains a lower distance value (more accurate) compared to TRILL across all scaling ratios, indicating superior trajectory recovery accuracy. During the experiments, we also found that the average training time per epoch ranges from 4.2s to 22.4s (scaling from 20% to 100% of full training data) which is still comparable with the best baseline. The training time can satisfy the common requirement in company services since this model is only for offline applications of trajectory recovery. These findings suggest that DiffMove exhibits better scalability and accuracy to variations in different data scales, making it a promising solution for trajectory recovery tasks across diverse datasets and scaling scenarios.

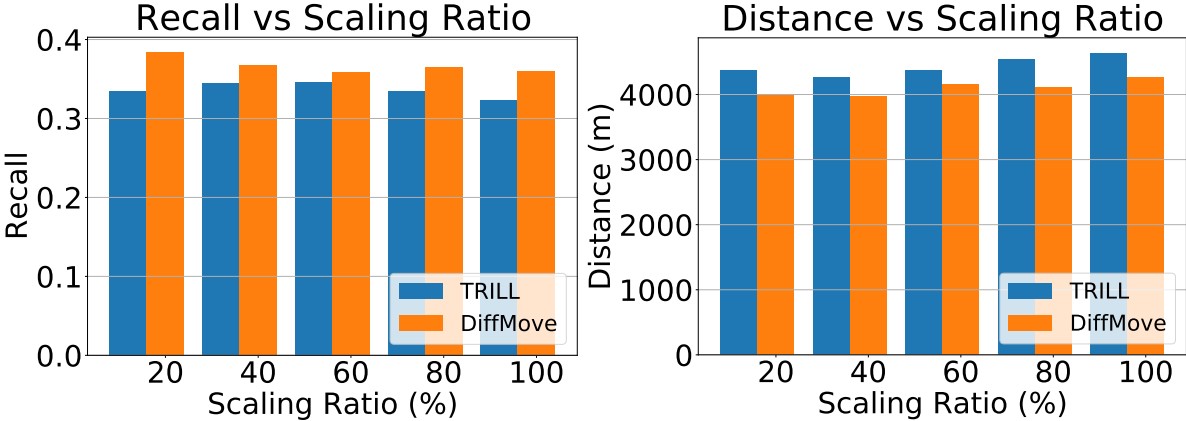

*Figure 9.* Recall vs Sample%              *Figure 10.* Distance vs Sample%

