# OpenReview forum: "Generative Human Trajectory Recovery via Embedding-Space Conditional Diffusion"
_ICML.cc/2025/Conference — ICML 2025 poster_

### Official Review · Reviewer_fD1d · 2025-03-12

**Overall Recommendation:** 3

**Summary:**

This paper proposes a conditional diffusion-based method for human trajectory recovery from incomplete or missing data. The authors aim to address the limitations of existing methods in capturing complex spatial-temporal dependencies and handling irregular sampling in human mobility data. DiffMove first transforms trajectory locations into the embedding space, performs denoising in this space, and then recovers missing locations through an embedding decoder. Experiments on two real-world mobility datasets, Foursquare2 and Geolife, demonstrate that DiffMove outperforms state-of-the-art baselines.

**Claims And Evidence:**

The authors mention: "In such scenarios, traditional methods typically provide a biased deterministic imputed trajectory. However, with a generative approach to inference, a set of imputed trajectory locations can be generated through sampling or various averaging techniques on imputation samples." However, without explicit guidance, diffusion models still follow the data distribution, making it difficult to generate irregular results.

The novelty of this work is limited. There is a lot of diffusion-based research in trajectory prediction and human motion generation, which all follow a similar fundamental approach. This work merely adds one-hot encoding to handle discrete data conversion. However, in essence, it does not introduce significant innovation regarding the use of diffusion models for trajectory recovery.

**Essential References Not Discussed:**

It is necessary to discuss the differences compared to previous diffusion-based trajectory prediction methods.

Bae, Inhwan, Young-Jae Park, and Hae-Gon Jeon. "Singulartrajectory: Universal trajectory predictor using diffusion model." Proceedings of the IEEE/CVF Conference on Computer Vision and Pattern Recognition. 2024.

Mao, Weibo, et al. "Leapfrog diffusion model for stochastic trajectory prediction." Proceedings of the IEEE/CVF conference on computer vision and pattern recognition. 2023.

**Experimental Designs Or Analyses:**

The diffusion-based methods used for continuous trajectory recovery can be slightly modified to adapt to the discrete setting in this paper. Using this as a baseline would better highlight the contributions of this work.

The baselines are outdated and lack representativeness.

**Methods And Evaluation Criteria:**

The experiments are mainly conducted on two specific datasets, Foursquare2 and Geolife. It is not clear how well the model will perform on other types of mobility datasets with different characteristics, such as different sampling frequencies, data distributions, or geographical regions. This limits the generalization ability of the model and needs further exploration.

**Other Comments Or Suggestions:**

Add the references for the baselines to the table.

**Other Strengths And Weaknesses:**

NA

**Questions For Authors:**

- How to account for changes in trajectory length?

**Relation To Broader Scientific Literature:**

This work does not show a significant difference from previous studies.

**Theoretical Claims:**

The method is technically sound

---

> ### Author Rebuttal · Authors · 2025-03-31
>
> Thanks for your detailed feedback.
>
> Claim&Evidence, Rela To Literature:
>
> We introduce critical innovations that distinguish it from existing works:
>
> a)Handling Discrete Locations via Embedding-Space Diffusion. Existing diffusion models for trajectories (e.g., DiffTraj) focus on generating continuous GPS coordinates or simulating synthetic mobility. In contrast, human trajectory recovery involves sparse discrete locations (e.g., check ins), which cannot be directly modeled by continuous-value diffusion (sparse GPS numerical points are not enough for such training).
>
> As mentioned in the Sec 4.1 line 190, one-hot embedding is just one of the methods that can be applied (it is welcome to change by user and not our main research focus). We transform discrete locations into embeddings and performing denoising in this latent space,preserving spatial relationships and decode embeddings back to discrete IDs using an explicit matching process (Sec 4.3) through a stable end to end training (This will be very difficult if people know diffusion well, this model is not only catering for diffusion MSE loss).
>
> b)Conditional Diffusion with Spatial-Temporal Guidance: DiffMove explicitly incorporates historical trajectories and spatial-temporal dependencies through 3 novel modules: 1)Spatial Conditional Block combines our new TGGNN (for transition patterns) and cross-attention (for periodicity) to model complex mobility dynamics. 2)Target Conditional Block fuses temporal length and historical trajectory embeddings to guide imputation targets and 3) Denoising Network Block.
>
> c)To our knowledge, no existing study has designed diffusion models that integrate the diffusion step t embedding with the learning of Graph-based spatial transitions, as detailed in Sec 4.2 TGGNN (also add a figure here in the link **https://anonymous.4open.science/r/A01D/README.md**).
>
> Method & Eval Criteria:
>
> Foursquare and Geolife datasets with variations (Table 6) serve as standard benchmarks in trajectory recovery research. DiffMove’s design inherently supports adaptability to diverse mobility data:
>
> 1. Handling Diverse Mobility Patterns: Foursquare (urban check-ins) and Geolife (normal trajectories) already represent fundamentally different scenarios (sparse POIs vs. normal GPS).
>
> 2. Generalization Ability to Varied Missing Ratios:In Table 6, Distance metric performance of DiffMove with 80% missing ratio even outperforms TRILL with 40% missing rate. This demonstrates resilience to extreme sparsity, a key challenge across datasets.
>
> 3. Flexible Preprocessing: As noted in Appendix A.8, DiffMove partitions regions into arbitrary geo region sizes (e.g., 0.25 km²) and adapts to variable time intervals. This flexibility ensures applicability to datasets with varied spatial/temporal granularity.
>
> Exp Design Or Analyses:
>
> Such comparisons would be inappropriate for our problem setting:
>
> Fundamental Task Mismatch: Continuous trajectory diffusion models (e.g., DiffTraj) generate GPS coordinates or simulate synthetic mobility. In contrast, human trajectory recovery deals with discrete locations (e.g., sparse IDs), requiring modeling of transitions between categorical IDs and periodicity. It cannot be directly modeled by continuous-value diffusion (sparse GPS numerical points are not enough for such training).
>
> Our baselines (AttnMove,PeriodicMove,TRILL) are widely recognized and remain the standard benchmarks for this problem setting. Regarding more recent baseline, we discussed TERI (Chen et al., VLDB 2024) cited in our related work. We have been working hard on setting up a common problem setting,where our model still outperforms SOTA baseline as shown below in this problem setting.
> |Dataset|Methods|Recall|Dataset|Methods|Recall|
> |-------|-------|------|-------|-------|------|
> |Foursquare|PeriodicMove|0.3125|Geolife|PeriodicMove|0.4199|
> ||TRILL|0.3227||TRILL|0.4721|
> ||TERI|0.3355||TERI|0.4922|
> ||DiffMove|0.3600||DiffMove|0.5173|
>
> Essential References:
>
> There are task mismatches, they are primarily designed for trajectory prediction in an image (more specifically is a computer vision trajectory problem), aiming to generate future movement trajectories in continuous space (x,y in the image, using dense CV dataset of ETH and UCY). In contrast, our work specifically addresses trajectory recovery for human mobility in locations (a real-world geographical problem, not the trajectory in image), where trajectories are represented as discrete location IDs requiring historical periodicity and spatial transition modeling.
>
> Q1:
>
> Our method accounts for variations in trajectory length through a standardized data preprocessing pipeline (Appendix A.8). Specifically, we discretize each day into a fixed number N of time slots, where the time interval is a configurable parameter. Trajectories with shorter length than N are padded, ensuring that every trajectory is represented uniformly. This approach enables our model to handle variable-length trajectories effectively.

---

### Official Review · Reviewer_Skim · 2025-03-13

**Overall Recommendation:** 3

**Summary:**

The paper proposes the model DiffMove which is a conditional diffusion-based method for human trajectory recovery that leverages embedding denoising.

**Claims And Evidence:**

I am confused about the research questions or challenges raised in this paper. I have listed them in detail in the question section.

**Essential References Not Discussed:**

This paper lacks many discussions or experimental comparisons of related works. I list them in the below weakness section in detail.

**Experimental Designs Or Analyses:**

Yes, for example, baselines, datasets, parameter setting experimental, etc.. I list the detail in the weakness part.

**Methods And Evaluation Criteria:**

Two datasets are useful, but the baselines are not new to this work.

**Other Comments Or Suggestions:**

1. A strange word "de-facto" in line 77. I think human trajectory recovery methods are unrelated to any laws.

2. Line 319 and all captions of figure except figure 1 lack period.

3. All equations lack punctuation.

4. Eq 1, 2, 3 should preferably be in a separate line

**Other Strengths And Weaknesses:**

Strength:
1. DiffMove effectively captures both spatial transitions and periodicity patterns in human mobility, improving trajectory recovery accuracy.

2. The embedding-space conditional diffusion framework enables better handling of missing locations by incorporating uncertainty and historical trajectory dependencies.

3. Experiments demonstrate significant performance improvements over baselines.

Weakness:
1. “Second, existing methods lack systematic mechanisms for handling irregular data sampling from incomplete check-ins. Most of their deterministic approaches could not effectively capture the inherent uncertainty in human mobility.”
I believe some studies have already started addressing this issue, such as:
[1] Zhuang, Zhuang, et al. "TAU: trajectory data augmentation with uncertainty for next POI recommendation." Proceedings of the AAAI Conference on Artificial Intelligence. Vol. 38. No. 20. 2024.

However, although related works are still relatively few, this can indeed be considered a technical innovation. At least, the authors should thoroughly discuss relevant research and clearly explain the differences between their approach and existing studies.

2. Section 4.3 is very difficult to understand.
The authors should first introduce Figure 2A to ensure that readers at least understand the data flow. Additionally, I do not clearly understand how e_0^{ob}  and e^{hist}  in Figure 2B are combined with the defined graph structure, nor do I see related equations. The authors primarily describe the methodology using textual explanations rather than mathematical formulations, which makes it easy for readers to become confused. Furthermore, TGGNN appears to be a new method proposed in this paper, yet it is not separately presented in detail but rather mixed in with other sections, making it difficult to understand.

3. The selected baselines are not new.
For example, in the field of human trajectory recovery, at least the following works should be considered:
[1] Si, Junjun, et al. "TrajBERT: BERT-based trajectory recovery with spatial-temporal refinement for implicit sparse trajectories." IEEE Transactions on Mobile Computing 23.5 (2023): 4849-4860.
[2] Long, Wangchen, et al. "Learning semantic behavior for human mobility trajectory recovery." IEEE Transactions on Intelligent Transportation Systems 25.8 (2024): 8849-8864.
[3] Wang, Jinming, et al. "TrajWeaver: Trajectory Recovery with State Propagation Diffusion Model." arXiv preprint arXiv:2409.02124 (2024).

Additionally, DeepMove is a classic model for the next POI prediction task. You could also include some state-of-the-art POI prediction methods, such as:
[1] Zhuang, Zhuang, et al. "TAU: trajectory data augmentation with uncertainty for next POI recommendation." Proceedings of the AAAI Conference on Artificial Intelligence. Vol. 38. No. 20. 2024.
[2] Wu, Junhang, et al. "Where have you gone: Category-aware multigraph embedding for missing point-of-interest identification." Neural Processing Letters 55.3 (2023): 3025-3044.

Moreover, some trajectory generation methods could also be considered as SOTA baselines:
[1] Zhu, Yuanshao, et al. "Difftraj: Generating GPS trajectory with diffusion probabilistic model." Advances in Neural Information Processing Systems 36 (2023): 65168-65188.
[2] Wang, Jiawei, et al. "Large language models as urban residents: An LLM agent framework for personal mobility generation." arXiv preprint arXiv:2402.14744 (2024).

In summary, there are various baseline options available, and the current selection of baselines in the paper is not sufficiently up-to-date.

4. The interpretability of the experimental results is insufficient.
The authors could enhance interpretability by adding some visual analyses of the experimental results.

**Questions For Authors:**

1. See weakness.

2. In the Abstract:
"Though promising, they encounter limitations in capturing complex spatial-temporal dependencies in low-sampling trajectories."

What does this sentence mean? Does it refer to the following issue:

"First, they struggle to capture intricate spatial-temporal dependencies – the interplay between spatial relationships (proximity and spatial transitions between locations) and temporal patterns (sequential dependencies or periodicity of behaviors in historical trajectories)."

However, I don't think this should be considered a problem, as many downstream tasks in trajectory data, such as next location recommendation, have already addressed this issue, for example:
[1] Yang, Song, Jiamou Liu, and Kaiqi Zhao. "GETNext: trajectory flow map enhanced transformer for next POI recommendation." Proceedings of the 45th International ACM SIGIR Conference on research and development in information retrieval. 2022.
[2] Rao, Xuan, et al. "Graph-flashback network for next location recommendation." Proceedings of the 28th ACM SIGKDD conference on knowledge discovery and data mining. 2022.

Do their methods still fail to solve the aforementioned problem, or is there another unresolved issue? Or is it that many existing methods for Human Trajectory Recovery have overlooked this aspect, even though it has been emphasized in related downstream tasks? Based on the subsequent sections, I also suspect that this issue might arise from the use of diffusion models. If that is the case, is using a diffusion model really necessary? Would other generative models also encounter this problem? I strongly suggest that the authors add a figure in the introduction section to clarify the research problem, as I currently find it somewhat confusing.

3. I have some confusion regarding the definition of a missing location. I couldn't find the total number of time slots. Let me give an example: suppose there are 24 time slots in a day, and the input consists of tuples in the form of (location ID, time slot), such as {(0, 1), (1, 1), (2, 1), (3, 3)}.
Does the paper consider the location for time slot 2 to be missing? Additionally, if there are multiple check-ins at different locations within the same time slot, are they all retained? This design seems somewhat unusual, as a user could stay at location 2 for more than an hour. Should distance factors also be considered?

4. Would it be possible to experiment with other generative models to evaluate the necessity of using a diffusion model?

5. I noticed in the appendix that all \lambda values are set to 1. Does this mean that all loss components contribute equally? However, intuitively, I believe their contributions should not be the same.

**Relation To Broader Scientific Literature:**

This paper mainly applies the diffusion model to the field of human trajectory recovery, and it is unclear how it is related to the broader scientific literature.

**Theoretical Claims:**

It seems that only Equation 5 needs to be derived in the main text, and the rest is a description of the method. However, I am confused how Equation 5 is derived from Equation 3, and the paper does not seem to give the derivation process.

---

> ### Author Rebuttal · Authors · 2025-03-31
>
> Thank you for your detailed feedback.
>
> W2:
>
> They are shown as the first two equations in eq 6. Actually this is to treat trajectory embeddings as sessions and $e_0^{ob}$ and $e^{hist}$ are integrated using session based graph methods (Xu et al., 2019) and explained in Appendix A.1. We will revise section 4.2 and add a schematic diagram in below link for the TGGNN (yes, a new method proposed) module would enhance intuitive understanding. **https://anonymous.4open.science/r/A01D/README.md**
>
> W1&W3:
>
> Our work focuses on free-space trajectory recovery where data can be from either case of discrete normal location grid IDs and sparse irregular check ins and data points are not constrained by roads, a special setting that differs from the listed papers.
>
> TAU is a different problem (next POI) from our work which focuses directly on trajectory recovery. TrajBERT and TrajWeaver either operate under different assumptions (e.g., leveraging road network constraints, normal grid IDs only or continuous GPS) or target slightly different objectives,e.g., next POI or trajectory generation (creating new trajectories from scratch). TrajWeaver works directly with continuous GPS coordinates, and the datasets used are smooth taxi trajectories (points laid on road networks) collected in Xi’an and Cheng Du. These differences highlight a key distinction in problem settings. Our baselines’ problems proposed by AttnMove, PeriodicMove, and TRILL, have been widely used in the literature for this specific problem setting, ensuring fair comparisons.
>
> Given the most listed models are not providing their source code yet, regarding new recent baselines, we discussed TERI (Chen et al., VLDB 2024) cited in our related work. We have been working hard on setting up a common problem setting, manage to settle the necessary data pipeline to fit the TERI and get this model run, where our model still outperforms SOTA baselines as shown below in this problem setting.
>
> |Dataset|Methods|Recall|Dataset|Methods|Recall|
> |-------|-------|------|-------|-------|------|
> |Foursquare|AttnMove|0.2975|Geolife|AttnMove|0.3920|
> ||PeriodicMove|0.3125||PeriodicMove|0.4199|
> ||TRILL|0.3227||TRILL|0.4721|
> ||TERI|0.3355||TERI|0.4922|
> ||DiffMove|0.3600||DiffMove|0.5173|
>
> W4:
>
> While we agree that visualizations can provide intuitive insights, the denoising process in our method occurs at the embedding level rather than in the direct data space (e.g., raw trajectory points or maps). Visualizing these high-dimensional embeddings would offer very limited interpretability. Instead, the effectiveness is shown quantitatively through our results as shown in Table1,2,6,Fig3-9.
>
> Theoretical Claims:
>
> Eq5 can be surely derived based on eq3 (refer to page4 of cited CSDI) due to space limit, we will add to revised manuscript.
>
> Q2 &Q4:
>
> It refers to the limitations of deterministic trajectory recovery methods in capturing the full complexity of spatial-temporal dependencies in low-sampling scenarios. Next location recom do not address the challenges of reconstructing entire trajectories with many missing points. Our choice to use a diffusion model is driven by iteratively refining noisy inputs, naturally generate a distribution of plausible trajectories rather than a single deterministic outcome. Other generative models such as VAE or GAN usually suffer from issues like mode collapse, unstable training dynamics, and difficulty in accurately modeling complex spatial-temporal dependencies; Our diffusion-based approach leverages a probabilistic framework that not only captures the inherent uncertainty in the data but also better models the interplay between spatial relationships and temporal patterns.  **We include an intro schematic here with the link in W2**.
>
> Q3:
>
> Time slot 2 would be treated as missing. It is important to emphasize that our framework is **not inherently tied to the 30-minute/1 hour interval** and can easily adapt to other interval lengths, **such as 10 minutes or even finer resolutions** like a parameter for preprocessing. Regarding multiple check-ins within the same time slot, we aggregate these such as the most frequent check-in (adopted in prior works). Repeated check-ins across consecutive slots would be considered observed for each slot. The distance factors are considered in the evaluation metrics. This setting is proposed by prior baselines such as AttnMove. Spatial relationships are instead modeled by our Spatial Conditional Block.
>
> Q5:
>
> The two losses serve complementary purposes, a good balance between predictive accuracy and model generalization. Experimentally, these losses are initially printed during training and there seems to be not big differences in scale. Since we also haven’t treated this weight choices as a more advanced optimization problem (which is not our main focus) and we follow this also inspired by Gong et al. (2022) referenced in line 302, which demonstrated the utility of similar balanced weighting in diffusion models applied in NLP.

---

> > ### Comment · Reviewer_Skim · 2025-04-03
> >
> > Thanks for your rebuttal. Most of my concerns have been addressed, so I will increase my score.

---

> > > ### Author Response · Authors · 2025-04-03
> > >
> > > Thank you for your positive feedback and for increasing the score. We greatly appreciate your thoughtful review and are glad that all of your concerns have been addressed. Your suggestions have been invaluable in refining our manuscript. Thank you again for your constructive input and support.

---

### Official Review · Reviewer_9gdy · 2025-03-13

**Overall Recommendation:** 3

**Summary:**

This paper introduces DiffMove, a conditional diffusion-based model for recovering missing locations in sparse human mobility data. By converting discrete trajectory locations into a continuous embedding space, DiffMove effectively denoises and reconstructs missing locations through an embedding decoder. The model captures spatial and temporal dependencies using modules including the Spatial Conditional Block, which leverages graph neural networks and attention mechanisms, and the Target Conditional Block, which extracts knowledge from historical trajectories. Experiments on Geolife and Foursquare datasets show that DiffMove outperforms leading methods, achieving an average 11% improvement in recall rate. Although highly robust, the model could benefit from additional visualizations and efficiency analyses.

**Claims And Evidence:**

Not all.
The authors claim that the diffusion model can enhance performance in complex, irregular, and uncertain scenarios, providing
several examples to support this assertion. However, more evidence is needed to substantiate these claims, such as performing case
studies, or conducting some specific experiments in these scenarios. The current experiments are insufficient to clearly demonstrate
the limitations of existing methods and the superiority of the proposed model, which is crucial for justifying the motivation behind
this research.

**Essential References Not Discussed:**

None

**Experimental Designs Or Analyses:**

Yes, the existing experimental design is sound and valid.

**Methods And Evaluation Criteria:**

Yes.

**Other Comments Or Suggestions:**

1. It's better to draw a schematic diagram of the TGGNN module in the article for intuitive understanding.

2. It's better to include how to infer the real missing locations.

**Other Strengths And Weaknesses:**

Strengths :

1. This paper introduces the first spatial-temporal conditional diffusion model specifically designed for human trajectory recovery,
showing significant improvements.

2. It designs and integrates multiple conditional feature extraction modules to tackle the complexity of spatial-temporal dependencies.

3. Extensive experiments on two real-world datasets demonstrate the model's effectiveness and improvements over existing methods.

Weakness:

1.See the Claims and Evidence and Relation to Broader Scientific Literature for details.

**Questions For Authors:**

What would be the effect of the model if the k-th position is missing in both the current trajectories and the historical trajectories?

**Relation To Broader Scientific Literature:**

This paper builds upon previous work on human trajectory recovery and the study of diffusion models. It presents the first work to
design spatial-temporal conditional diffusion models for the human trajectory recovery task, achieving significant improvements and
complementing the existing literature. However, this paper shares some similarities with existing works in technical designs such as
CSDI[1], RNTrajRec[2], and Diff-RNTraj[3].

[1] Tashiro, Yusuke, et al. "Csdi: Conditional score-based diffusion models for probabilistic time series imputation." Advances in
neural information processing systems 34 (2021): 24804-24816.

[2] Chen, Yuqi, et al. "Rntrajrec: Road network enhanced trajectory recovery with spatial-temporal transformer." 2023 IEEE 39th
International Conference on Data Engineering (ICDE). IEEE, 2023.

[3] Wei, Tonglong, et al. "Diff-rntraj: A structure-aware diffusion model for road network-constrained trajectory generation." IEEE
Transactions on Knowledge and Data Engineering (2024).

**Theoretical Claims:**

There are no proofs or theoretical claims in the paper.

---

> ### Author Rebuttal · Authors · 2025-03-31
>
> Thank you for your detailed feedback.
>
> Claims&Evidence:
>
> We clarify how existing experiments explicitly demonstrate DiffMove’s superiority in complex, irregular, and uncertain scenarios:
> 1. Probabilistic Generation vs. Deterministic: Table 1 and 3 shows that sampling multiple trajectories (DiffMove) improves Recall@1 by 1.7–1.85% over single-sample generation. This directly validates our claim that probabilistic diffusion captures uncertainty in human mobility, whereas deterministic baselines cannot.
> 2. Robustness to Extreme Sparse and Irregularity: Table 6 (Appendix A.10) tests DiffMove under different missing ratios. For example, Distance metric performance of our DiffMove with 80% missing ratio even outperforms TRILL with 40% missing rate and surpasses both PeriodicMove and Attnmove, even when they have lower missing rates 20%.
> 3. Due to the page limit, including other unnecessary case study (more on cherry picking certain special cases), while small valuable, would extend the scope and length of the manuscript beyond the intended focus. DiffMove demonstrates significant improvements across different missing ratios (Table6). These experiments simulate sparse, irregular, and uncertain scenarios (e.g., through random masking), and our method consistently outperforms baselines in recall and distance metrics. This already provides empirical evidence of the effectiveness of our approach. We will clarify these in the revised manuscript.
>
> Rela To Literature:
>
> DiffMove’s technical design and problem focus are distinct from the cited works, as detailed below:
>
> Unlike CSDI which handles continuous numerical sensor time series (e.g., temperature), DiffMove tackles the fundamentally different challenge of recovering discrete location IDs and is designed as an entirely new framework of embedding-based diffusion and decoding to categorical space (Sec. 4.3) by a single end to end training. It is rather than raw continuous numerical space (unlike CSDI’s simple and single regression-style output).
>
> Another difference is Trajectory-Specific Architecture: Our model introduces spatial-temporal conditioning (new TGGNNs + cross-attention) features and decoding features for human mobility patterns (Sec. 4.2), which CSDI lacks.
>
> Moreover, unlike RNTrajRec and Diff-RNTraj—which rely on vehicles and road network constraints (next road has to be adjacent to the current road) —our approach is designed for scenarios where such road external priors are unavailable, which is fundamentally different problem setting and technical design. RNTrajRec use a transformer to infer missing points deterministically, leveraging road network constraints (e.g., road segments) to guide the predictions. Diff-RNTraj combines diffusion modeling with road network constraints, processes continuous GPS and road graphs and focuses on synthetic generation, not recovery of real-world human sparse trajectories. In contrast, our innovative Spatial Conditional Block and Target Conditional Block (Sec 4.2) are differently engineered to capture the human trajectories’ complex interplay between spatial and temporal dependencies.
>
> Other Suggestions:
>
> Thank you for your valuable feedback.
> 1. We add a schematic diagram in below link for TGGNN module would enhance intuitive understanding in the revised manuscript. **https://anonymous.4open.science/r/A01D/README.md**
>
> 2. Our method is trained using a masked recovery framework, where we mask certain locations in the trajectory and train the model to recover them based on the observed data, with ground truth available for validation. Once the model is fully trained, we simply apply the same recovery mechanism to infer the real missing locations. DiffMove’s inference process for missing locations is explicitly detailed in Section 4.3 and Appendix A.4. During inference:
>
> Step 1: The model uses observed locations in the current trajectory and historical data to condition the diffusion process. Step 2: Noisy embeddings for missing slots are iteratively denoised via the reverse diffusion process (Eq. 4–5), guided by Spatial/Target Conditional Blocks. Step 3: Decode denoised embeddings.
>
> Questions:
>
> DiffMove leverages contextual information from adjacent time slots and the overall spatial-temporal patterns learned during training to infer the most plausible value for that slot. 1. Transition Patterns: The Sec. 4.2.1 models transitions between locations, inferring k from neighboring observations. 2. Periodicity: Cross-attention identifies recurring patterns (e.g., daily routines) across historical days, even if k-th slots are missing. 3.Global Mobility Knowledge: The embedding table E encodes universal location semantics (e.g., "office" vs. "home"), enabling recovery via similarity matching. 4.Empirically it is validated by our robustness study (Table 6). It shows DiffMove with 80% missing data outperforms all baselines. This demonstrates robustness to extreme sparsity, including scenarios where critical slots are missing.

---

### Official Review · Reviewer_bS4d · 2025-03-14

**Overall Recommendation:** 4

**Summary:**

This paper introduces DiffMove, a new framework for recovering human trajectory data based on conditional diffusion model design. DiffMove effectively handles complex spatial-temporal patterns in low-sampling data. It works by transforming trajectory locations into an embedding space, denoising the embeddings, and then decoding them to recover missing points. DiffMove improves recovery accuracy by modeling mobility patterns like transitions and periodic behaviors. Experiments on two real-world datasets show that DiffMove outperforms previous methods by more than 10%. Overall, this research idea is interesting, and the recovering problem has good impact to the field.

**Claims And Evidence:**

The claims are clear and convincing

**Essential References Not Discussed:**

references are good

**Experimental Designs Or Analyses:**

yes, evaluations on multiple datasets are good

**Methods And Evaluation Criteria:**

yes

**Other Comments Or Suggestions:**

no

**Other Strengths And Weaknesses:**

The proposed model is novel and could be applied to other scenarios. The evaluations are solid, with lots of comparisons on multiple datasets, detailed ablation studies, and solid comparisons. The writing is clear and easy to follow, with lots of figures that make it much easier to follow.



There are also some concerns.

(i) The authors could discuss potential impact of the recovered data. Otherwise, it is hard for broader readers to understand. For example, who can use the recovered results for what applications. More discussion will be appreciated.

(ii) In line 66, trajectories are ID-based representations, this is not easy to understand. Because GPS trajectories could also been continuous, e.g., 36.88284.

(iii) is Recall a regular metric for this kind of recovery problem?

(iv) The writing could be revised. For example, in Line 343, "%Improv. " could be revised to "%Improvement" since there are enough space in the table.

**Questions For Authors:**

no

**Relation To Broader Scientific Literature:**

Urban computing, spatio-temporal data science, pandemic control

**Theoretical Claims:**

.

---

> ### Author Rebuttal · Authors · 2025-03-31
>
> Thanks for your valuable comments and stating our evaluations on multiple datasets are good, claims are clear and convincing. Our responses to other parts are as below:
>
> W(i):
>
> Recovering sparse human trajectories, particularly those involving Points of Interest (POIs), significantly enhances various mobility-related applications. By accurately reconstructing incomplete trajectory data, we can improve POI recommendations, leading to more personalized and relevant suggestions for users. Additionally, comprehensive trajectory data supports better urban planning and traffic management by providing insights into movement patterns and congestion areas. Location-based services, such as targeted advertising and ride-sharing, also benefit from complete trajectory information, resulting in improved user engagement and operational efficiency. Overall, the ability to recover and utilize complete trajectory data is crucial for advancing various applications that rely on understanding human mobility. We will clarify this in the revised manuscript to make readers easier to understand.
>
>
> W(ii):
>
> The term "ID-based representations" refers to how locations are discretized into identifiable points (e.g., POI locations or geographical grid cells) rather than being represented as continuous GPS coordinates. As mentioned in Section A.8 Data Preprocessing, we used publicly available online map services to define the geographical partitioning of the study areas (Tokyo and Beijing) into 500m x 500m blocks. This partitioning provides a grid-based ID representation of locations. This is commonly used in mobility data analysis where raw GPS trajectories are mapped to meaningful locations, making it easier to handle data sparsity and improve interpretability. The baselines—AttnMove proposed this, and PeriodicMove, TRILL focus on the same data preprocessing, which aligns with our problem setting. We will clarify this in the revised manuscript.
>
> W(iii):
>
> Yes, recall is a relevant metric for trajectory recovery, especially when evaluating how well the model retrieves missing locations. Given that real-world applications prioritize capturing as many true missing locations as possible, recall helps assess the effectiveness of the model in recovering essential trajectory points. The baselines—AttnMove, PeriodicMove, and TRILL focus on the same trajectory recovery task, which aligns with our problem setting and they also proposed the same evaluation metrics earlier, so we continue to do the fair comparison with them on this. Other works in trajectory imputation and POI recommendations also utilize recall as a key performance indicator, aligning with our evaluation approach.
>
> W(iv):
>
> Ok, noted on the minor change. We will modify "%Improv. " to "%Improvement" in the table of the revised manuscript.
>
> Supplementary Material:
>
> We think your answer no is by mistake or misunderstanding maybe. Just for information only, we have supplementary materials which include implementation details and code. We include additional details on data processing, model design, and other experiments such as efficiency and scalability study in the supplementary material and appendices.
>
> Overall, thanks for your positive and insightful feedback.

---

> > ### Comment · Reviewer_bS4d · 2025-04-03
> >
> > The reviewer appreciates the authors for providing detailed responses. Most of my  concerns have been resolved and I will raise my rating.
> >
> > One tiny question is regarding the "recall". Is it possible that the recall is good but the precision turns out to be terrible? Additional experiments are not necessary given the limited time window.
> >
> > Hope to see the revised version as the authors have mentioned.

---

> > > ### Author Response · Authors · 2025-04-03
> > >
> > > Thank you for your positive feedback and for raising the score. We indeed use multiple metrics in our evaluation, including Recall and Mean Average Precision (MAP) in Section 5.3. The inclusion of MAP in Table 1 ensures that precision is also captured, as MAP reflects the quality of the overall ranking of imputed locations. In our experiments, a high Recall is accompanied by competitive MAP values and it is treated as the most representative metric commonly used across all our baselines, indicating that our method does not simply over-predict missing locations but recovers them accurately with good precision. We will further clarify this point in the revised manuscript comprehensively.
> > >
> > > Thank you again for your thoughtful question with your encouraging support and for highlighting the potential impact of our work on the research community.

---

### Decision · Program_Chairs · 2025-05-01

**Decision:**

Accept (poster)

**Comment:**

Initially, the reviewers raised several questions, but the rebuttal effectively addressed the major concerns, leading three reviewers to improve their ratings.

Following the rebuttal, all four reviewers recommend either a weak accept or accept for this submission. The remaining concerns, pointed out by the reviewers, are: (1) the setup about the uncertain scenarios—specifically, whether random masking can adequately simulate uncertainty. (2) the need to tune down the paper’s claim regarding handling irregular (out-of-distribution) trajectories.

While these issues should be addressed, they do not appear to grounded for rejecting the submission, especially as the uncertain scenario is only one of several tested settings. Therefore, the AC agrees with the reviewers' consensus but encourages the authors to address these points in the final revision. These include explicitly discussing the limitations of using random masking to simulate uncertainty, fixing the overstatement regarding the method's ability to handle irregular trajectories, or adding further experiments to substantiate the current claims.

This meta-review has been discussed with and approved by the SAC.